# Effect of Adjuvant and Palliative Chemotherapy in Large Cell Neuroendocrine Carcinoma of the Lung: A Systematic Review and Meta-Analysis

**DOI:** 10.3390/cancers13235948

**Published:** 2021-11-26

**Authors:** Hao Chen, Masashi Ishihara, Nobuyuki Horita, Hiroki Kazahari, Ryusuke Ochiai, Shigeru Tanzawa, Takeshi Honda, Yasuko Ichikawa, Kiyotaka Watanabe, Nobuhiko Seki

**Affiliations:** 1Division of Oncology, Department of Internal Medicine, Teikyo University School of Medicine, Tokyo 173-8606, Japan; m.ishihara@med.teikyo-u.ac.jp (M.I.); kazahari0309@med.teikyo-u.ac.jp (H.K.); ryo7132003@med.teikyo-u.ac.jp (R.O.); s.tanzawa@med.teikyo-u.ac.jp (S.T.); thonda@med.teikyo-u.ac.jp (T.H.); icchi@med.teikyo-u.ac.jp (Y.I.); kiyowata@med.teikyo-u.ac.jp (K.W.); 2Department of Pulmonology, Yokohama City University Graduate School of Medicine, Yokohama 236-0027, Japan; horitano@yokohama-cu.ac.jp

**Keywords:** large cell neuroendocrine carcinoma of the lung, chemotherapy, regimen, small cell lung cancer, non-small cell lung cancer

## Abstract

**Simple Summary:**

Adjuvant chemotherapy revealed a better outcome than surgery only, but there was no statistical difference in patients with stage IA. The small cell lung cancer regimen (SCLC) was frequently selected in adjuvant chemotherapy. The SCLC regimen showed better survival than the non-SCLC regimen as palliative chemotherapy at the endpoint of the odds ratio of mortality after two years.

**Abstract:**

Background: Pulmonary large cell neuroendocrine carcinoma (LCNEC) is a rare subset of lung carcinoma with poor overall survival. Methods: A systematic review following a meta-analysis of studies was performed to identify the effect of different selections of chemotherapy in LCNEC. Articles providing overall survival data for adjuvant chemotherapy or palliative chemotherapy for LCNEC were eligible. The odds ratio (OR) of mortality at one or two years after chemotherapy was evaluated. Results: A total of 16 reports were finally included in the quantitative synthesis, involving a total of 5916 LCNEC patients. Adjuvant chemotherapy was administered to 1303 patients, and palliative chemotherapy was administered to 313 patients using either a small cell lung cancer (SCLC) or a non-small cell lung cancer (NSCLC) regimen. The OR for adjuvant chemotherapy was 0.73 (95% confidence interval (CI): 0.59 to 0.89, *p* = 0.002). The SCLC regimen showed an OR of 0.52 (95% CI: 0.11 to 2.38, *p* = 0.40) after one year, and 0.32 (95% CI: 0.11 to 0.89, *p* = 0.03) after two years, compared with the NSCLC regimen. Conclusions: Adjuvant chemotherapy for pulmonary large cell neuroendocrine carcinoma improved the outcome after surgery. The SCLC regimen showed better survival than the NSCLC regimen as palliative chemotherapy.

## 1. Introduction

Pulmonary large cell neuroendocrine carcinoma (LCNEC) is a rare subset of lung carcinoma with poor overall survival; it accounts for less than 3% of all lung malignancies. According to the 2015 World Health Organization criteria, high-grade neuroendocrine carcinomas of the lung are classified into two categories: LCNEC and small cell lung carcinoma (SCLC) [1]. Whereas typical cases of LCNEC are morphologically distinct from SCLC, the differentiation between LCNEC and SCLC can be challenging in some cases [2]. Because of the difficulties in diagnosing LCNEC, and its rarity, the optimal systemic treatment has not been established [3]. In the American Society of Clinical Oncology (ASCO) guidelines, either platinum plus etoposide or the same regimen as other patients with non-squamous carcinoma is advised for LCNEC [4].

There are limited data on the usage and selection of chemotherapy in early-stage and advanced-stage LCNEC. In early-stage LCNEC, the mainstay of treatment for the node-negative disease is surgical resection, and optimal adjuvant treatment strategies are not well defined. Adjuvant chemotherapy regimens are associated with varying efficacy in preventing tumor recurrence [5]. In advanced-stage LCNEC, SCLC regimens are more commonly used based on limited retrospective analyses, [6] and several studies showed that the genomic profile of LCNEC corresponds closely with that of SCLC [7,8]. Recently, it was suggested that there is an increase in overall survival (OS) in LCNEC patients when NSCLC regimens are adopted, especially gemcitabine–platinum rather than pemetrexed–platinum and etoposide–platinum (SCLC regimens) [9].

The standard chemotherapy for LCNEC is unclear. This study was designed to examine the effect of adjuvant and palliative chemotherapy for LCNEC. The effect of adjuvant chemotherapy was checked by comparing patients who underwent surgery plus adjuvant chemotherapy and surgery-alone patients. The effect of palliative chemotherapy was checked by comparing the effects of the SCLC and NSCLC regimens.

## 2. Materials and Methods

### 2.1. Study Overview

The protocol of this systematic review and meta-analysis followed the Preferred Reporting Items for Systematic Reviews and Meta-Analyses (PRISMA) statement and was registered on the website of the University Hospital Medical Information Network Clinical Trials Registration (UMIN000044028) [10,11]. Institutional review board approval was not required because of the nature of this study. The PRISMA checklist is shown in Appendix A.

### 2.2. Study Search

Four major online databases, namely, PubMed, Web of Science, Cochrane, and Embase, were searched. The following formula was applied for PubMed: (large cell neuroendocrine carcinoma) OR (LCNEC) OR (high-grade neuroendocrine carcinomas) AND (chemotherapy). Two review authors (MI and HC) independently screened the titles and abstracts and carefully evaluated the full text to select eligible articles. In cases of discrepancy, they reached a consensus through discussion. Review articles, as well as original articles, were hand searched (MI and HC) for additional research papers that met the inclusion criteria.

### 2.3. Inclusion and Exclusion Criteria

Full articles, brief reports, and conference abstracts published in any language that provided data for OS after chemotherapy for LCNEC were included. To be included, a study had to include: (1) patients with pulmonary LCNEC, (2) data outlining mortality two years after chemotherapy, and (3) numbers of patients in each group.

Exclusion criteria were as follows: (1) single-arm study, (2) SCLC was included and data for LCNEC could not be separated, and (3) staging information was not available.

### 2.4. Quality Assessment

The target population was patients with LCNEC. Commonly used pathological criteria were accepted, along with WHO 2015 criteria [12,13]. The SCLC regimen was platinum and etoposide or irinotecan; the NSCLC regimen was different combinations of platinum with gemcitabine, pemetrexed, docetaxel, paclitaxel, or vinorelbine. 

### 2.5. Risk of Bias

Two reviewers independently assessed the methodological quality of selected studies using the Newcastle–Ottawa quality assessment, evaluating the quality of observational studies [14]. Disagreement between reviewers was discussed, and agreement was reached by consensus.

### 2.6. Outcomes

Typically, five-year OS is the gold standard in adjuvant chemotherapy; the effect of adjuvant chemotherapy was checked by comparing the ORs of mortality in patients who underwent adjuvant chemotherapy or surgery alone at five years. In palliative chemotherapy, most of the studies reported OS after two years. The different effects of the SCLC and NSCLC regimens were checked by ORs of mortality one and two years after the different regimens. 

### 2.7. Data Extraction

Two review authors, MI and HC, independently extracted data, including the name of the first author, publication year, publication country, staging, and regimen used in the treatment.

### 2.8. Statistics

All analyses were performed in Review Manager ver. 5.3 (Cochrane Collaboration, Oxford, UK). Figures illustrated using Review Manager were adjusted as necessary. Odds ratios were determined by calculating the numbers of surviving patients one or two years after chemotherapy. Heterogeneity evaluated with the I^2^ statistic was interpreted as follows: I^2^  =  0% indicates no heterogeneity, 0%  <  I^2^  <  25% indicates the least heterogeneity, 25%  ≤  I^2^  <  50% indicates mild heterogeneity, 50%  ≤  I^2^  <  75% indicates moderate heterogeneity, and 75%  ≤  I^2^ indicates strong heterogeneity [15]. A *p*-value of <0.05 was considered significant.

## 3. Results

### 3.1. Study Search and Study Characteristics

We identified 2872 articles, including 2870 articles through database search and 2 articles by hand search. There were 1056, 1658, and 100 articles left after removing duplication, screening, and full-article reading, respectively (Appendix A). We finally included 15 reports in quantitative synthesis, including a total of 6457 LCNEC patients. All of them were retrospective studies, and one study checked the effect of adjuvant chemotherapy by using propensity scores in matched pairs. The Newcastle–Ottawa Scale ranged from five to seven stars.

All of them were written in English. There were eight studies that compared the effect of adjuvant chemotherapy (Table 1) [16,17,18,19,20,21,22,23]. Among the nine reports, two were from Germany and USA, and there was one each from China, Canada, Japan, and Korea. Adjuvant chemotherapy was conducted in 1303 of 5603 LCNEC patients who underwent an operation. Detailed information on adjuvant chemotherapy was described in five studies. The SCLC regimen was more frequently selected than NSCLC in adjuvant chemotherapy, and one study used the SCLC regimen of adjuvant chemotherapy for all patients.

Six studies compared the effect of chemotherapy in different regimens. A total of 131 cases of SCLC regimens and 182 cases of NSCLC regimens were conducted in the treatment of LCNEC (Table 2) [6,9,16,24,25,26]. Platinum plus etoposide or irinotecan was used in the SCLC regimen; different combinations of platinum were used in the NSCLC regimen. There were four studies that revealed detailed combinations of chemotherapy in the NSCLC regimen, and gemcitabine was the most frequently used. Only one study focused on the effect of different regimens in the treatment of stage IV LCNEC. 

### 3.2. Effect of Adjuvant Chemotherapy

In 1458 cases of adjuvant chemotherapy of LCNEC, the mortality of patients was evaluated after adjuvant chemotherapy or surgery only, after five years. The OR of adjuvant chemotherapy was 0.65 (95% confidence interval (CI): 0.53 to 0.80, *p* < 0.01; I^2^ = 40%, *p* for heterogeneity = 0.11) (Figure 1). Subgroup analysis of adjuvant chemotherapy in patients with stages I, IA, and IB was conducted separately. The OR in patients with stage I was 0.68 (95% CI: 0.48–0.95, *p* = 0.02; I^2^ = 59%, *p* for heterogeneity = 0.06) (Figure 2a), whereas ORs were 0.88 (95% CI: 0.71–1.08, *p* = 0.22; I^2^ = 0%, *p* for heterogeneity = 0.75) (Figure 2b) and 0.49 (95% CI: 0.32–0.73, *p* < 0.01; I^2^ = 44%, *p* for heterogeneity = 0.17) (Figure 2c) in patients with stages IA and IB. 

### 3.3. Effect of Different Regimens in Chemotherapy

The OR of mortality was counted at one and two years after chemotherapy by the SCLC or NSCLC regimen separately. At the endpoint of one year after chemotherapy, there was no difference in the OR of SCLC and NSCLC regimens. The OR was 0.52 (95% CI: 0.11 to 2.38, *p* = 0.40; I^2^ = 71%, *p* for heterogeneity = 0.004) (Figure 3a). However, the SCLC regimen showed a better effect than NSCLC two years after chemotherapy. The OR of the SCLC regimen was 0.32 (95% CI: 0.11 to 0.89, *p* = 0.03; I^2^ = 46%, *p* for heterogeneity = 0.10) (Figure 3b).

There was no obvious publication bias in the meta-analysis of adjuvant chemotherapy and palliative chemotherapy because no obvious funnel plot asymmetry was observed (Appendix A).

## 4. Discussion

This study evaluated the effect of adjuvant chemotherapy in 6114 cases of LCNEC that underwent resection and revealed that chemotherapy improved survival five years after chemotherapy. There was no standard regimen for adjuvant chemotherapy; the SCLC regimen was selected more frequently than the NSCLC regimen. Adjuvant chemotherapy was proved to be effective at all stages of analysis, but there was no statistical difference in patients with stage IA in the subgroup analysis. In palliative chemotherapy, the SCLC regimen showed a tendency for better survival than the NSCLC regimen after one year and was proven to have a better outcome than the NSCLC regimen after two years by comparing 301 chemotherapy patients. Generally, the SCLC regimen seemed to be a better choice than the NSCLC regimen in palliative chemotherapy of LCNEC. 

The selection of the regimen for LCNEC was difficult. Moderate heterogeneity (I^2^ = 71%) was observed in the OR one year after palliative chemotherapy. Chemotherapy in stage Ⅳ LCNEC reported by Derks et al. played an important role in heterogeneity. However, the trend did not seem to be the same as in other papers limited to stage III–IV patients. The mean age of patients in Derks’s study seemed older than other studies. The SCLC regimen might be less effective than the NSCLC regimen in older patients with stage Ⅳ LCNEC because of its adverse effects. Some of the cases had been treated with chemoradiation therapy. Chemotherapy showed a limited effect in the treatment of LCNEC, and a combination of radiotherapy might be a potential selection for the treatment of LCNEC. 

Molecular subtype-based evidence for the selection of the regimen was reported recently. Patients with LCNEC tumors that carry a wild-type RB1 gene or express the RB1 protein fare better with NSCLC-GEM/TAX treatment than with SCLC-PE chemotherapy. [27] Moreover, there was no consensus in the selection of chemotherapy for LCNEC. LCNEC is one of the rare cancers, and large-scale prospective clinical research is difficult. Unlike other pathological types of lung cancer, limited achievement has been obtained in the treatment of LCNEC. The expression and prognostic impact of programmed cell death ligand 1 (PD-L1) in LCNEC were assessed in several studies, and the high frequency of PD-L1 expression could support the use of PD-L1 antibody in the treatment of LCNEC. [28,29] Evidence of the efficiency of immune checkpoint inhibitors was only observed in case reports [30,31,32]. There were also several case reports on the efficiency of chemotherapy by epidermal growth factor receptor tyrosine kinase inhibitors (EGFR-TKI) [33,34]. 

Several limitations of this study must be considered when interpreting the results. Given the nature of the rare disease of LCNEC, there were no prospective studies enrolled. Second, the effects of chemotherapy were evaluated by the OR of mortality one, two, or five years after the treatment. Hazard ratios of PFS and OS were more useful in appraising the effect of chemotherapy. Third, there was only one study that adjusted the backgrounds of patients enrolled in this study; there was a substantial risk of selection bias due to the nature of the observational study.

## 5. Conclusions

Adjuvant chemotherapy revealed a better outcome than surgery only, and the SCLC regimen was frequently selected. In palliative chemotherapy, the SCLC regimen showed a better effect than the NSCLC regimen.

## Figures and Tables

**Figure 1 cancers-13-05948-f001:**
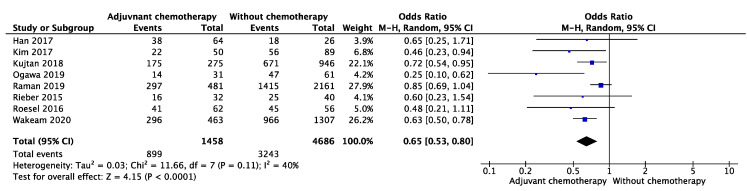
Effect of adjuvant chemotherapy after five years in patients of all stages.

**Figure 2 cancers-13-05948-f002:**
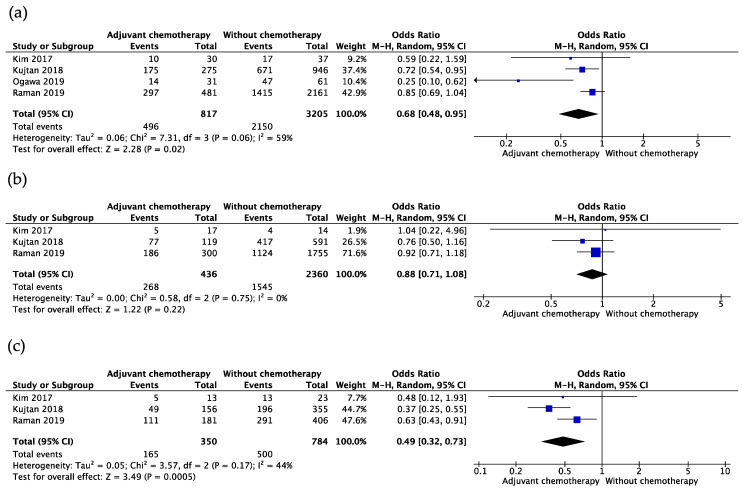
Subgroup analysis of adjuvant chemotherapy in early stage. (**a**) The odds ratio of adjuvant chemotherapy in patients with stage IA; (**b**) odds ratio of adjuvant chemotherapy in patients with stage IB; (**c**) odds ratio of adjuvant chemotherapy in patients with stage I.

**Figure 3 cancers-13-05948-f003:**
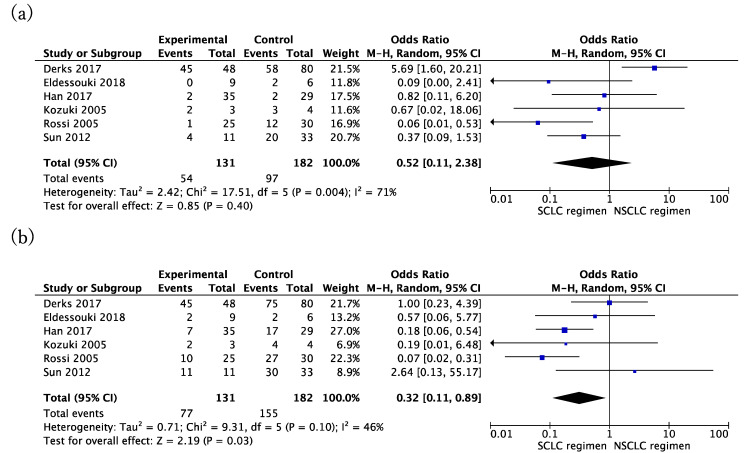
Different effects of chemotherapy by the SCLC or the NSCLC regimen. (**a**) The odds ratio of the small cell regimen versus the non-small cell regimen after one year; (**b**) odds ratio of the small cell regimen versus the non-small cell regimen after two years.

**Table 1 cancers-13-05948-t001:** Background and characteristics of studies of adjuvant chemotherapy.

Author	Country	No. of Cases	No. Chemo	Nature	Age (y)	Stage	Regimen	NOS
Han 2017	China	90	64	Retro	62	I–III	35 SCLC regimen; 29 NSCLC regimen	6
Kim 2017	Korea	139	50	Retro	65	I–IV	Paclitaxel or docetaxel plus platinum	6
Kujtan 2018	USA	1232	275	Retro	66	I	The regimen was not described	5
Ogawa 2019	Japan	92	31	Retro	68	I–III	20 SCLC regimen; 11 NSCLC regimen	6
Raman 2019	USA	1755	300	Retro	66	I	The regimen was not described; 54 cases of chemoradiation, 46 cases of radiation	5
Rieber 2015	Germany	66	32	Retro	63	I–IV	16 SCLC regimen; 16 NSCLC regimen	6
Rossel 2016	Germany	118	62	Retro	64	I–II	The regimen was not described	5
Wakeam 2020	Canada	1017	463	Retro	67	I–II	The regimen was not described; 33 cases of radiation enrolled	5

Retro: retrospective; SCLC: small cell lung cancer; NSCLC: non-small cell lung cancer; NOS: Newcastle–Ottawa Scale.

**Table 2 cancers-13-05948-t002:** Background and characteristics of studies of chemotherapy with a different regimen.

Author	Country	Total Cases	SCLC Regimen	Stage	Age (y)	Details of the NSCLC Regimen	NOS
Derks 2017	Netherlands	128	48	IV	65	GEM 46; PEM 20; PTX 7; DOC 6; VNR 1	6
Eldessouki 2018	USA	15	9	III–IV	55	Detailed information not available	5
Han 2017	China	64	35	I–III	62	Platinum with GEM 9; VNR 8; PEM 7; DOC 5	6
Kozuki 2005	Japan	7	3	III–IV	64	Platinum with DOC 2; GEM 2; VNR 1	6
Rossi 2005	France	55	25	I–III	65	Detailed information not available	5
Sun 2012	Korea	45	11	I–IV	64	Platinum with GEM 9; PTX 4; PEM 2; DOC 2; VNR 2	6

SCLC: small cell lung cancer; NSCLC: non-small cell lung cancer; NOS: Newcastle–Ottawa Scale; GEM: gemcitabine; PEM: pemetrexed, PTX: paclitaxel, DOC: docetaxel; VNR: vinorelbine.

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
