# Peer review of "Effect of Adjuvant and Palliative Chemotherapy in Large Cell Neuroendocrine Carcinoma of the Lung: A Systematic Review and Meta-Analysis"

_cancers, 2021, doi:10.3390/cancers13235948_

Round 1

Reviewer 1 Report

This is a well written systematic review on the effect of chemotherapy treatments in Large Cell Neuroendocrine Lung Cancer, a rare tumor with poor prognosis and very limited therapeutic options.

The authors have  clearly defined the objectives and background of the study and made a thorough online search of the published papers in the field.  The content is valuable for LCNEC management until a more effective therapy becomes available.

The paper performs an exhaustive bibliographic examination of an understudied type of cancer with poor prognosis and very limited (if any) effective options. [This is worthwhile].

The objective of the study is clearly defined (identify the effects of different selections of chemotherapy regimens in LCNEC). So the authors stick to the objective, develop their study with the data available (surely less than desirable, but there is a worrying lack of knowledge about this type of lung cancer) and draw a conclusion accordingly.

The conclusion is sound and beneficial for patients with the disease. Authors were required just to revise minor specific points, so I hope they can be responded without any particular concern. It is therefore recommended to be published in this journal.

Major Critiques:

None

Minor Points:

Line 19: LCNEC, instead of LENEC

Line 110: identify author SK

Lines 140,150 (abbreviations to the tables): revise editing

Line 163: Insert “in patients of” before of all stages

Author Response

Dear reviewer

We wish to express our strong appreciation to the reviewers for their insightful comments on our paper. We feel the comments have helped us significantly improve the paper.

Minor comment 1

Line 19: LCNEC, instead of LENEC

Response: We appreciate the reviewer's comment on this point. We have changed LCNEC instead of LENEC.

Minor comment 2

Line 110: identify author SK

Response: We appreciate the reviewer's comment on this point. We edited our manuscript as “Two review authors, MI and HC, independently extracted data…”.

Minor comment 3

Lines 140,150 (abbreviations to the tables): revise editing

Response: Response: We appreciate the reviewer's comment on this point. We revised abbreviations to make them at the bottom of the tables.

Minor comment 4

Line 163: Insert “in patients of” before of all stages

Response: Response: We appreciate the reviewer's comment on this point. We modified content as “Figure 1. Effect of adjuvant chemotherapy after five years in patients of all stages.”

Thanks again for your treasure comments on this article, which made it more readable.

Best Regards.

Reviewer 2 Report

Your research has been interesting. I think the conclusions are convincing.
However, I have a few questions.

The paper you refer to in Table 2, "Kozuki 2005", is not found in the references, and the paper numbered 24 seems to be a different paper.
Also, shouldn't the reference numbers be [6,9,15,23,25] instead of [9,15,23-25]?

In the discussion, you suggest that the effect of chemotherapy may be different in stage 4, based on Derks' article.
However, the trend does not seem to be the same as in other papers limited to stage 3-4 patients.
Can you discuss why there is such a difference between a paper on stage 4 only and a paper on stage 3-4? Are there any other differences other than stage?

I would think that radiotherapy could be considered for localised disease. Some of the cases in the article you cite also seem to have been treated with radiotherapy, so is it still difficult to suggest anything about radiotherapy?

Author Response

Dear reviewer

We wish to express our strong appreciation to the reviewers for their insightful comments on our paper. We feel the comments have helped us significantly improve the paper.

Minor comment 1

The paper you refer to in Table 2, "Kozuki 2005", is not found in the references, and the paper numbered 24 seems to be a different paper.

Also, shouldn't the reference numbers be [6,9,15,23,25] instead of [9,15,23-25]?

Response: We appreciate the reviewer's comment on this point. We added the reference of “Kozuki 2005”.  

Minor comment 2

In the discussion, you suggest that the effect of chemotherapy may be different in stage 4, based on Derks' article. However, the trend does not seem to be the same as in other papers limited to stage 3-4 patients. Can you discuss why there is such a difference between a paper on stage 4 only and a paper on stage 3-4? Are there any other differences other than stage?

Response:We appreciate the reviewer's comment on this point.

We modified our manuscript as “However, the trend did not seem to be the same as in other papers limited to stage III-IV patients. The mean age of patients in Derks’s study seemed older than other studies. SCLC regimen might be less effective than NSCLC regimen in elder patients of LCNEC in stage â…£ because of its adverse effects.”

Minor comment 3

I would think that radiotherapy could be considered for localised disease. Some of the cases in the article you cite also seem to have been treated with radiotherapy, so is it still difficult to suggest anything about radiotherapy?

Response:

We appreciate the reviewer's comment on this point. We added this content in discussion section as follows: “Some of the cases had been treated with chemoradiation therapy. Chemotherapy showed a limited effect in the treatment of LCNEC and a combination of radiotherapy might be a potential selection in the treatment of LCNEC.”

Thank you for your useful suggestion, which made our manuscript more readable.

Best regards

Round 2

Reviewer 2 Report

I think the additional observations you have made are appropriate.
You have also corrected the errors in the references.